# Targeting Mitochondrial Dynamics Proteins for the Development of Therapies for Cardiovascular Diseases

**DOI:** 10.3390/ijms232314741

**Published:** 2022-11-25

**Authors:** Alexander V. Blagov, Sergey Kozlov, Tatiana Blokhina, Vasily N. Sukhorukov, Aleksander N. Orekhov

**Affiliations:** 1Laboratory of Angiopathology, Institute of General Pathology and Pathophysiology, 8 Baltiiskaya Street, 125315 Moscow, Russia; 2National Medical Research Centre of Cardiology, Ministry of Health of the Russian Federation, 15A, 3-rd Cherepkovskaya Street, 121552 Moscow, Russia

**Keywords:** cardiovascular disease, mitochondria, mitochondrial dynamic

## Abstract

Cardiovascular diseases are one of the leading causes of death worldwide. The identification of new pathogenetic targets contributes to more efficient development of new types of drugs for the treatment of cardiovascular diseases. This review highlights the problem of mitochondrial dynamics disorders, in the context of cardiovascular diseases. A change in the normal function of mitochondrial dynamics proteins is one of the reasons for the development of the pathological state of cardiomyocytes. Based on this, therapeutic targeting of these proteins may be a promising strategy in the development of cardiac drugs. Here we will consider changes for each process of mitochondrial dynamics in cardiovascular diseases: fission and fusion of mitochondria, mitophagy, mitochondrial transport and biogenesis, and also analyze the prospects of the considered protein targets based on existing drug developments.

## 1. Introduction

Cardiovascular disease (CVD) is a general term for a number of related pathologies commonly defined as coronary heart disease (CHD), cerebrovascular disease, peripheral arterial disease, rheumatic and congenital heart disease, and venous thromboembolism [1]. The etiology of CVD is complex and includes metabolic abnormalities, genetic changes, abnormal protein function, and other factors. Globally, CVD accounts for 31% of deaths, most of which are due to coronary artery disease and cerebrovascular accidents. According to the World Health Organization (WHO), cardiovascular diseases are also the leading cause of death worldwide, causing 17.9 million deaths each year [2]. The incidence of CVD worldwide is projected to increase, as the prevalence of CVD risk factors increases in previously low-risk countries. Currently, 80% of cardiovascular deaths occur in developing countries. The burden of cardiovascular disease is further increased as it is considered the most costly disease, even ahead of Alzheimer’s and diabetes. Although age-adjusted rates and acute CVD mortality have declined over time, reflecting advances in diagnosis and treatment over the past couple of decades, the risk of CVD remains high, with an estimated 50% risk by age 45, overall. The incidence increases significantly with age, with some differences between the sexes, as the incidence is higher in males at a younger age [1].

Cardiovascular disease is multifactorial; some risk factors can be changed, and some (age, heredity and male sex) cannot be changed. One modifiable risk factor for cardiovascular disease is a diet high in saturated fat and sodium. Other risk factors are tobacco smoke, high blood cholesterol, high blood pressure, physical inactivity, obesity and overweight, and diabetes. Stress and excessive alcohol consumption may also contribute to the risk of cardiovascular disease [3]. The most dangerous complication of cardiovascular disease is death, and despite numerous discoveries made in recent decades, cardiovascular disease remains one of the leading causes of death worldwide. Other complications, such as the need for longer hospital stays, disability, and increased treatment costs, are significant and under the spotlight as they are expected to continue to rise in the coming decades [4].

Modern methods of treating cardiovascular diseases mainly include traditional pharmacotherapy and surgery [5]. Although the above methods alleviate the symptoms of the disease and reduce mortality, both methods have certain disadvantages. Traditional drugs are less invasive but can cause damage to the liver, kidneys, and other organs, as well as other side effects [6]. Despite its excellent efficacy, the clinical application of cardiac surgery is always constrained by the complexity of procedures and the possibility of postoperative complications [7]. Thus, there is an urgent and unmet need to develop a new, convenient and effective approach to the treatment of cardiovascular diseases. The discovery of new markers of CVD pathogenesis is a preliminary step towards the development of new cardiac drugs. Consideration of changes in the dynamics of mitochondria in CVD look promising as pathogenetic changes, since mitochondria are essential organelles for maintaining the vital activity of cardiomyocytes. Mitochondria are the “powerhouses” of cells; they are involved in ATP formation, calcium homeostasis, response to oxidative stress, and apoptosis. Thus, maintaining mitochondrial function is critical for cells. The data obtained in recent years have shown a significant role of mitochondria in the pathogenesis of cardiovascular diseases, which makes them attractive therapeutic targets for combating CVD [8].

## 2. The Role of Mitochondria in Cardiac Functioning

### 2.1. ATP Production

The heart constantly consumes a large amount of energy, mainly for the purposes of contraction and transport of ions. In cardiomyocytes, mitochondria occupy approximately one third of the cell volume, reflecting the high energy requirements of these cells. Mitochondria produce over 95% of the ATP in the myocardium. To provide a constant supply of energy and at the same time adapt to stimuli, mitochondria transform various available substrates into ATP. This substrate plasticity adjusts to environmental changes, such as the availability of oxygen and nutrients in the blood, as well as rapid changes in workload, such as during exercise. Fatty acids are the preferred substrate, accounting for 60–90% of myocardial energy supply [9]. Once fatty acids are taken from the bloodstream, they are transported to the mitochondria for fatty acid β-oxidation. In this sequence of reactions, long-chain fatty acids are cleaved to acetyl-CoA, which then enters the tricarboxylic acid (TCA) cycle. The electron transporters, FADH 2 and NADH, are produced during the CTC, which transfers electrons to the electron transport chain (ETC). This, in turn, directs protons into the intermembrane space to activate ATP synthase when oxygen is consumed. In contrast, glucose is first converted in the cytosol to pyruvate, a process called glycolysis. The mitochondrial pyruvate dehydrogenase complex converts pyruvate to acetyl-CoA, which in turn feeds the TCA and ETC. Fatty acid and glucose metabolism regulate each other in a negative feedback loop known as the Randle cycle, which ensures that available resources are used optimally at all times [10]. Other energy substrates such as ketones, amino acids, and lactate can also be oxidized to maintain the CCA and produce ATP. Their contribution to the general energy of the myocardium at rest is insignificant, but becomes important under various stress conditions [9].

### 2.2. Calcium Homeostasis

Mitochondria are also responsible for the regulation of ion transport, and their many ion channels and exchangers play an important role in cell protection. The most important of these are the selective channels K^+^ and Ca^2+^, Na^+^/H^+^ and K^+^/H^+^ exchanger. Proper function of these ion channels is required to maintain the mitochondrial membrane potential, which supports ATP production and redox regulation [9]. Among these ions, Ca^2+^ plays a central role in the regulation of many cellular functions, including metabolism and the effects of transcription-modifying enzymes [11]. In cardiomyocytes, cytosolic calcium is released during contraction, which leads to subsequent removal of calcium from the cytosol during relaxation. Because they contain several different ion channels and transporters, mitochondria are able to sense cytosolic calcium signals and mediate the release of calcium from the cytosol into the mitochondrial matrix. Ca^2+^ is able to easily diffuse through the outer mitochondrial membrane (OMM) through voltage-dependent anion channels (VDAC) [12]. In contrast, the mitochondrial inner membrane (IMM) is impermeable to ions, highlighting the need to identify mitochondrial Ca^2+^ transport proteins. The mitochondrial calcium uniporter (MCU) on the IMM is the central mediator of mitochondrial calcium influx, and is tightly controlled by two regulatory proteins: mitochondrial calcium uptake 1 and 2 (MICU1 and MICU2). MICU1/2 prevents excessive accumulation of Ca^2+^ inside the mitochondrial matrix, maintaining the driving force of the proton [13].

### 2.3. Lipid Synthesis

Mitochondria also play a critical role in lipid homeostasis. Most lipids are synthesized in the endoplasmic reticulum (ER); however, some major mitochondrial membrane components are synthesized in the IMM from imported lipids. Distearoylphosphatidylserine (PS) and phosphatidic acid (PA) are imported from the ER membrane into the OMM via membrane contact sites called mitochondria-associated ER membranes. The lipid precursors are then transported across the intermembrane space by protein complexes such as ubiquitin-specific peptidase or the PRELID1-TRIAP1 complex [14]. Upon reaching IMM, phosphatidylserine decarboxylase converts PS to phosphatidylethanolanime (PE); meanwhile, PA goes through a series of reactions and is converted into the phospholipid cardiolipin. Both phospholipids are especially abundant in the IMM and are essential for the structure of the cristae. PE plays a role in the folding of membrane proteins, but more importantly, it has been found to be required for efficient mitochondrial respiration [15].

## 3. Processes of Mitochondrial Dynamics in the Heart Muscle

Mitochondria are not stationary organelles: they can fuse with each other, divide, move between cell compartments, break down, and be synthesized with the help of the coordinated work of many regulatory proteins. Mitochondrial dynamics is vital for maintaining the normal functioning of cellular homeostasis. It regulates the quantity, quality, distribution and localization of mitochondria in the cell, which is extremely important for maintaining its energy needs.

### 3.1. Mitochondrial Fusion

The process of mitochondrial fusion involves the fusing of both the outer (OMM) and inner mitochondrial membrane (IMM) of the two merging organelles. The main proteins that control this process in mammalian cells are encoded in the nucleus and belong to the GTPase family of dynamins. These include mitofusin 1 (Mfn1), mitofusin 2 (Mfn2), and optic nerve atrophy protein 1 (Opa1). Mfn1 and Mfn2 mediate outer membrane fusion, while Opa1 is required for inner mitochondrial membrane fusion. The Mfn1/2 pair with their homologues is located on the OMM of another participating mitochondrion, then GTP is hydrolyzed, creating a force impact that is crucial for membrane fusion [16]. Two possible OMM merger models are being considered. The first suggests that the GTP-binding domain of HR2 promotes membrane docking, during the formation of dimers of antiparallel helical mitofusins that bring opposite mitochondrial membranes closer together at a distance of ~10 nm from each other; after which, the HR1 domain, which has a conserved amphipathic helix, interacts with the lipid membrane bringing the OMM closer, which disrupts the structure of their lipid bilayer. The second model suggests that GTP binding triggers the oligomerization of Mfns dimers and subsequent hydrolysis of GTP, which leads to a conformational change bringing the membranes closer together [17]. Conversely, Opa1 GTPase is located on the IMM, regulates mitochondrial fusion, and is also involved in cristae remodeling in response to energy stress or mitochondrial damage, thereby maintaining the integrity and function of the inner mitochondrial membranes [18]. Given the enormous amount of energy required for myocardial contractility, having an intact mitochondrial fusion network is critical to maintaining proper heart function. In addition, MFN2 is involved in several other physiological functions such as the modulation of energy processes, the interaction of the endoplasmic reticulum and mitochondria, and the regulation of mitophagy; this is the process by which mitochondria are taken up by autophagosomes and delivered to lysosomes for degradation [19].

### 3.2. Mitochondrial Fission

Mitochondrial fission is mainly regulated by dynamin-related protein 1 (Drp1). After activation, Drp1 translocates from the cytosol to mitochondria, which is facilitated by specific adapter proteins located in the outer mitochondrial membranes, such as mitochondrial fission protein 1 (Fis1), mitochondrial fission protein 1 (Mdv1), and mitochondrial fission factor (Mff) [20]. After being recruited to the outer mitochondrial membranes, Drp1 undergoes GTP-dependent oligomerization with the formation of a helical ring. This process involves each of four adjacent mitochondrial membranes being directed into a narrow lumen, resulting in a reorganization that has been shown to involve proteins such as dynamin 2, endophilin 1, and sorting nexin 9 (SNX9) [21]. The resulting ring, consisting of Drp1 molecules, initiates division if additional signal stimuli appear at the sites of potential mitochondrial cleavage, which include the interaction of Drp1 with actin filaments and the endoplasmic reticulum (ER). ER mitochondria, before division, initiate its initial contraction. Actin filaments promote assembly of the mitochondrial division complex and also activate Drp1, which increases the mechanical force of Drp1 ring contraction. The protein dynamin-2 (Dyn2) was also found, which forms an additional constriction at the end of the process of mitochondrial division [22]. Fission proteins also play a crucial role in adjusting mitochondrial metabolism and are required to meet the energy needs of the heart muscle [10].

### 3.3. Mitophagy

Mitophagy is a special case of autophagy, the process of cellular self-degradation of misfolded or aggregated proteins, as well as damaged organelles. Autophagy plays an important role in the regulation of mammalian heart development from the embryonic stage. It is known that knockout of genes encoding regulatory proteins of autophagy, leads to impaired development of the heart [23]. In addition, the role of impaired autophagy in the development of cardiovascular diseases is noted. Since one of the key factors in the development of CVD is arterial hypertension associated with a disruption of the activity of a number of hormonal proteins that regulate blood pressure, as well as the process of autophagy, there is a relationship between these processes [23]. Mitophagy is a cellular process that specifically identifies and eliminates damaged and dysfunctional mitochondria [19]. Mitophagy can occur through parkin-dependent or parkin-independent mechanisms. Parkin-independent types of mitophagy include nutrient-deficient mitophagy and stress-related mitophagy. In both cases, sequestration of mitochondria occurs through the formation of a mitophagosome, in contrast to the Parkin-dependent pathway, during which mitochondria are localized for degradation in multivisicular bodies. Parkin-dependent mitophagy occurs most frequently and is associated with a state of normal cellular homeostasis [24]. In parkin-mediated mitophagy, kinase 1, induced by phosphatase and the PINK1 protein, recruits parkin to the outer mitochondrial membranes. This may involve direct phosphorylation of parkin or other indirect mechanisms such as PINK1 phosphorylation of Mfn2. After activation, parkin ubiquitinates various targets, allowing them to interact with mitophagy adaptors, for example, p62/sequestosome 1 [25]. Then, the interaction of the attracted sequestosome 1 protein-1 light chain 3 associated with microtubules (LC3) mediates the uptake of damaged mitochondria by autophagosomes [26]. Finally, sequestered mitochondria are digested by lysosomes, fusing with autophagosomes to form so-called “autolysosomes”. Parkin activity is also counteracted by several mechanisms, including inhibition of its translocation to outer mitochondrial membranes by p53 and deubiquitination of its targets by USP15 and USP30 on mitochondrial outer membrane proteins [27]. Mitophagy can also occur through parkin-independent mechanisms, such as mediated by Bcl2-like protein 13 (Bcl2-L-13), mitochondrial receptor Nip3-like protein X (NIX), cardiolipin, or FUN14, domain-containing protein 1 (Fundc-1). In addition to activating mitophagy, Fundc-1 is also able to regulate mitochondrial fusion and fission by interacting with Opa1 and Drp1, respectively, located on the outer mitochondrial membranes [27].

### 3.4. Mitochondrial Biogenesis

Mitochondria have their own DNA. However, mitochondrial DNA encodes only a few components of the electron transport chain complexes and 22 mitochondrial tRNAs and rRNAs [28]. The rest of the ETC components and proteins required for mitochondrial translation and other components, are synthesized in the cytoplasm based on nuclear genetic material. Thus, mitochondrial biogenesis requires the simultaneous and coordinated expression of nuclear and mitochondrial genes. The PPARγ 1 coactivator (PGC1α) is a key activator of transcription and a major regulator of mitochondrial biogenesis [29]. It regulates the process of mitochondrial biogenesis through the activation of several other transcription factors involved in the expression of nuclear and mitochondrial genes. Activation of transcription factors, nuclear respiratory factors 1 and 2 (NRF-1 and NRF-2) and ERR leads to the induction of mitochondrial transcription factor A (TFAM) [30]. TFAM interacts directly with the mitochondrial genome and, together with the mitochondrial transcription factor B2 (TFB2M), induces the transcription of mitochondrial genes. In addition, PGC-1α promotes mitochondrial fatty acid oxidation by acting as a coactivator of peroxisome proliferator-activated receptors α and δ (PPARα and PPAR δ), resulting in the expression of mitochondrial genes responsible for fatty acid β-oxidation [31]. Thus, PGC1α activation leads to an increase in mitochondrial mass and substrate oxidation. Mitochondrial biogenesis is a physiological response to an increased need for energy, which leads to an increase in the level of AMP: ADP/ATP. Thus, PGC1α activation leads to an increase in mitochondrial mass and substrate oxidation. Mitochondrial biogenesis is a physiological response to an increased need for energy, which leads to an increase in the level of AMP: ADP/ATP. Overexpression of PGC1α in the heart enhances the expression of metabolic regulators, including Krebs cycle enzymes, as well as components of the oxidative phosphorylation complex involved in the electron transport chain [32]. In addition, PGC1α activation leads to a decrease in cellular oxidative stress by increasing the expression of mitochondrial antioxidant enzymes such as superoxide dismutase [33]. The molecular pathway of PGC1α activation is associated with the action of AMP-activated protein kinase (AMPK), whose action is induced in response to an increase in the AMP/ATP ratio, and is associated with the phosphorylation of PGC1α at Thr 177 and Ser 538. As a result of phosphorylation, the concentration of PGC increases -1 α on the promoters of its target genes, which leads to the activation of their expression. In addition, AMPK increases the level of nicotinamide adenine dinucleotide (NAD+), which causes the activation of Sirtuin1, which in turn leads to the activation of PGC1α by deacetylation [32]. It is noted that PGC1α is involved in the regulation of other processes of mitochondrial dynamics, by controlling the expression of genes of the main regulatory proteins of mitochondrial fission, fusion, and mitophagy. Thus, the study [34] notes that activation of PGC1α leads to an increase in the expression of MFN2 and OPA1 and a decrease in the expression of DRP1 and FIS1, which enhances mitochondrial fusion and weakens mitochondrial division, thus maintaining the balance of these processes. It is also known that PGC1α is able to activate the mitophagy process, both directly through the activation of PINK1 expression, and indirectly through the ERR α-SIRT pathway [32]. The multiple roles of PGC1α in the regulation of mitochondrial dynamics are shown in Figure 1. The total scheme of mitochondrial dynamics in cardiomyocytes is shown in Figure 2.

## 4. Mitochondrial Dynamics Disorders in Cardiovascular Diseases

### 4.1. Mitophagy Disorders

Efficient clearance of dysfunctional mitochondria is important for quality control of the entire mitochondrial network inside the cell, and, accordingly, for ensuring the viability of cardiomyocytes [35]. It was found that in various CVDs, the mitophagy process is impaired, which leads to the development of other anomalies and the progression of the disease. Thus, in the study of human samples from patients with atherosclerosis, as well as in the evaluation of mouse models of atherosclerosis, a general decrease in the level of autophagy based on p62 and LC3-II markers was found [36]. At the same time, as a result of induced mitophagy, the administration of low-density lipoproteins (LDL) and melatonin prevented the development of the disease [37]. Most likely, a decrease in autophagy and mitophagy, in particular, contributes to the accumulation of dead cells in the walls of arteries, which leads to the formation of plaques; the exact mechanism of mitophagy disruption in atherosclerosis is not fully known [38]. A decrease in the level of mitophagy is also observed in heart failure, which leads to the development of oxidative stress and damage to mitochondrial proteins, lipids, and DNA [39]. Evidence of this is the decrease in PINK1 activity in patients with heart failure, as well as the development of heart failure shown in a mouse model with PINK1 deficiency [40]. In ischemic heart disease, changes in mitophagy are more complex. As a result of ischemia–reperfusion injury (IRI), the potential of the mitochondrial membrane decreases and the transition pore of mitochondrial permeability opens, which leads to stimulation of apoptosis and death of cardiomyocytes [41]. In this situation, mitophagy is enhanced, which at the beginning of IRI protects cells from death; however, with extensive damage, mitophagy can become excessive, which leads to further development of the disease [38]. At the same time, IRI can negatively affect mitophagy, reducing it [40]. The molecular mechanism of mitophagy modulation during IRI depends on the phosphorylation of the BNIP3 protein: phosphorylation at the S17 position promotes the binding of BNIP3 and LC3, which enhances mitophagy; and phosphorylation at positions Y18 and S13, leads to the opposite effect [40].

### 4.2. Mitochondrial Transport Disorders

In addition to mitophagy, impaired mitochondrial transport can also be one of the pathogenesis factors detected in CVD. Impaired mitochondrial transport causes an abnormal distribution of mitochondria in the cell, which for cells with a large need for ATP absorption, such as cardiomyocytes, can lead to disruption of cellular bioenergetics and the correct performance of a number of functions, which leads to the development of pathology. One of the key proteins of mitochondrial transport, Miro1, is known to be highly expressed in cardiomyocytes [42]. With the progression of a number of CVDs, a change in the properties of microtubules is noted through which, as is known, intracellular transport occurs, including the movement of mitochondria [43]. It is found that in heart failure, as a result of myocardial stiffness due to microtubule remodeling, there is a change in the location of subsarcolemmal mitochondria which ultimately leads to arrhythmogenesis [38].

### 4.3. Mitochondrial Biogenesis Disorders

Mitochondrial biogenesis is an important component of mitochondrial dynamics necessary for the restoration of the mitochondrial pool. For a number of CVDs, a disruption of mitochondrial biogenesis is noted. Thus, it is noted that with mitochondrial myopathies, there is an increase in mitochondrial biogenesis in cardiomyocytes [44]. This condition may be associated with a compensatory response to an energy deficit caused by mitochondrial dysfunction. However, compensation does not occur, as evidenced by the continued decline in ATP production. In a mouse model, it was noted that during cardiac hypertrophy, the content of mitochondria increases in proportion to the cell growth of hypertrophied cardiac tissue [45]. At the same time, the number of mitochondria decreases with cardiac decompensation [46]. Additionally, when examining samples of cardiac tissue from patients with hypertrophic cardiomyopathy and terminated stage of heart failure, a decrease in the level of mitochondrial biogenesis was found, based on a decrease in the level of mitochondrial DNA itself and its transcription regulators [47]. Thus, the exact mechanism of changes in mitochondrial biogenesis remains unclear and can vary greatly for different CVDs, and even individual stages of the same disease.

### 4.4. Mitochondrial Fission Disorders 

The fission of mitochondria is important for maintaining the required bioenergetic state of the cell, however, excessive mitochondrial division, which is often observed during stressful and pathological conditions, can have negative consequences. For a number of CVDs, an aberrant increase in mitochondrial fission was found. It has been proven that mitochondrial fission is enhanced both in physiological hypertrophy of the heart and in pathological hypertrophy, leading to chronic hypertension. In the first case, hypertrophy is reversible, while the second leads to the death of cardiomyocytes and the development of heart failure [48]. Increased division of mitochondria is an adaptive response to cardiac hypertrophy, which increases bioenergetics and promotes mitophagy of dysfunctional mitochondria. However, excessive division at a constant high hemodynamic load leads to harmful consequences for cardiomyocytes, including activation of apoptosis and necrosis [48]. In addition, excessive mitochondrial fission is considered one of the main factors leading to the death of cardiomyocytes in ischemia–reperfusion injury. Mitochondrial fission is enhanced during ischemia and further maintained at a high level during reperfusion, which leads to increased generation of reactive oxygen species; opening of the mitochondrial permeability transition pore and calcium load; as well as impaired endothelial barrier function, which ultimately leads to cardiac dysfunction [48]. The role of aberrant mitochondrial fission has also been identified in the pathogenesis of metabolic cardiomyopathy, which is a consequence of obesity and type 2 diabetes mellitus [49]. Metabolic cardiomyopathy develops as a result of increased lipid uptake; in addition to excessive load on mitochondria during increased fatty acid oxidation, some lipids, including ceramide, are able to initiate mitochondrial division [50]. The role of enhanced mitochondrial fission in CVD has been proven in many studies, when comparing the level of Drp1 protein expression in pathology and in the norm, as well as when assessing the effect of Drp1 gene knockout on the manifestation of symptoms of the disease in animal models [51].

### 4.5. Mitochondrial Fusion Disorders 

The fusion of mitochondria plays an important role in maintaining the proper functioning of the mitochondrial network in the cell. This process is inextricably linked with other processes of mitochondrial dynamics, in particular with mitochondrial fission. The correct balance of fission and fusion of mitochondria is extremely important for maintaining the energy state of cells. Additionally, if in general for CVD there is a tendency of increasing of the mitochondrial fission, then there is also a tendency of weakening of the mitochondrial fusion. Reduced expression of mitochondrial fusion regulatory proteins such as MFN1, MFN2, and OPAI has been demonstrated in animal and patient models for conditions of cardiac hypertrophy and heart failure [48]. The same conclusion was made for ischemia–reperfusion injury of the heart and metabolic cardiomyopathy [52]. Decreased mitochondrial fusion leads to accumulation of dysfunctional mitochondria, subsequent enhancement of mitophagy, and triggering of mitochondria-mediated cardiomyocyte apoptosis. These studies are summarized in Table 1.

A generalized scheme of disturbances in the processes of mitochondrial dynamics in CVD, is shown in Figure 3.

## 5. Analysis of Mitochondrial Dynamics Proteins as Therapeutic Targets 

Since the mechanism of disruption of mitochondrial dynamics in CVD is most accurately determined for the processes of mitochondrial fusion and fission, most of the therapeutic strategies for the treatment of CVD aimed at modeling mitochondrial dynamics, are focused on the regulatory proteins of mitochondrial fusion and fission: Drp1, OPAI, Mfnl, and Mfn2. There are also developments aimed at modulating the signals of mitophagy regulators Pink1/Parkin, but their focus may differ in the context of different CVDs.

### 5.1. Impact on Mitophagy Proteins 

Pink and Parkin are the main proteins initiating the initial phase of mitophagy. They are considered as potential targets for the treatment of a number of diseases associated with disruption of the mitophagy process, primarily Parkinson’s disease [53]. The drug liraglutide, which is used to treat type 2 diabetes mellitus, has shown a new potential for restoring myocardial function after a heart attack. Liraglutide increased SIRT1 expression, which triggered Parkin-dependent mitophagy, reduced oxidative stress, and inhibited mitochondria-dependent apoptosis [54]. Melatonin, used as a sedative, has been shown to be effective in improving clinical symptoms in animal models of atherosclerosis [55] and diabetic cardiomyopathy [56] through activation of mitophagy and inhibition of inflammatory pathways. The alkaloid berberine, used to treat diarrhea, promoted the restoration of cardiac function in heart failure by activating Pink 1/Parkin-mediated mitophagy [38]. Since, in an ischemia/reperfusion state, excessive and persistent mitophagy may contribute to further disease pathogenesis, inhibition of mitophagy at this stage may be a potential treatment strategy. Superoxide dismutase mimetics, such as TEMPOL and mitoTEMPOL, can be used as inhibitors of mitophagy in such a situation [38]. Thus, at the moment there are several drug compounds that have shown their effectiveness in the treatment of CVD in animal models by modulating mitophagy. These compounds are already registered drugs for the treatment of other diseases, which will allow repurposing of their use, reducing the amount of research needed and reducing their cost. At the same time, the lack of completed clinical trials does not yet allow us to assess the likelihood of such drugs being released.

### 5.2. Impact on Mitochondrial Fission Proteins 

The DRP1 protein is the main activator of mitochondrial fission. Since many CVDs are characterized by an aberrant increase in mitochondrial division, inhibition of DRP1 activity appears to be a promising strategy in CVD therapy. Thus, the developed mitochondrial fission inhibitor mdivi-1, which suppresses the GTPase activity of DRP1, protected the heart from ischemia/reperfusion and reduced the possibility of myocardial infarction, which was shown in an animal model [57]. However, long-term inhibition of Drp1 by exposure to mdivi-1 led to the development of a negative scenario, which included the accumulation of damaged mitochondria and impaired cardiac function in cardiac hypertrophy and diabetic cardiomyopathy [58,59,60]. Short P110 peptides that inhibit the binding of DRP1 to its receptor on mitochondria, Fis1, look like a promising development. The advantage of these peptides, in contrast to the mdivi-1 compound, is the ability to distinguish between the pathological and physiological activity of Drp1, which prevents the risk of side effects [61]. The drug exenatide, which is a GLP-1 peptide mimetic and used in the treatment of type 2 diabetes mellitus, is able to phosphorylate Drp1 at S637, which leads to inhibition of the mitochondrial localization of Drp1 [62,63]. In a study [64], it was shown that GLP-1 treatment of the VSMC A7r5 cell line, which was derived from fetal rat aorta, induced heart regeneration. However, in a phase IV clinical trial [65] after the administration of exenatide, no significant improvement in cardiac function was found in patients with type 2 diabetes mellitus who had left ventricular systolic dysfunction. The compound resveratrol, isolated from grape extract, had a cardioprotective effect in various animal models [62]. It is believed that the mechanism of its action is based on the activation of SIRT1, through which the expression of Drp1 is suppressed [62]. In a clinical study in a group of patients with coronary heart disease who had a myocardial infarction, resveratrol significantly improved endothelial function and protected against adverse hemorheological changes [66]. Thus, with regard to DRP1, there are a number of inhibitory compounds—both new, but with a high potential for use—and those that have passed clinical studies, including those with a positive therapeutic effect. The unambiguity of the direction of mitochondrial fission disorders, gives DRP1 an additional advantage as a target for the therapy of a wide range of CVDs. 

### 5.3. Impact on Mitochondrial Fusion Proteins 

Just as an increase in mitochondrial fission has been found in the pathogenesis of various CVDs, a weakening of mitochondrial fusion has also been recorded. Mitochondrial fusion regulatory proteins such as Mfnl, Mfn2 and OPA1 can be considered as the main targets of therapeutic intervention. To date, there are very few studies on targeting these proteins. One of the therapeutic compounds aimed at enhancing mitochondrial fusion can be the enzyme heme oxygenase-1 (HO1), which is able to cleave prooxidant heme to carbon monoxide, biliverdin, and iron [62,67]. In a mouse model of dilated cardiomyopathy, it was shown that overexpression of HO-1 increases the level of mitochondrial fusion through an increase in Mfn1/2 expression [68]. Melatonin is able to affect the Notch1/Mfn2 signaling pathway, which leads to an increase in Mfn2 expression [69]. A potentially promising option could be the use of a peptide with a sequence complementary to the HR1 region of the Mfn2 protein. This peptide showed the ability to increase mitochondrial fusion by changing the Mfn2 conformation [70]. However, this study was conducted in an in vitro model of neurodegenerative disease, and it is unclear what effects a similar study would have on a CVD model. Thus, the few developments currently focused on the use of the Mfn2 protein as a target, while OPA1 and Mfn1 are still neglected. Potential therapeutic strategies aimed at modulating mitochondrial dynamics for CVD therapy, are summarized in Table 2.

## 6. Discussions

Regarding the development of new drugs targeted to mitochondrial dynamics proteins, the determination of the dose and duration of administration of these drugs remains an important issue. As discussed in the previous section, prolonged administration of the Drp1 inhibitor had a negative effect on cardiac function. This remark applies to all regulatory proteins of mitochondrial dynamics. The processes of mitochondrial dynamics are in a delicate balance between each other, which ensures their coordinated work in maintaining the correct functioning of the mitochondrial network in the cell; any disruption of this balance, both in the direction of strengthening or weakening one of the processes, can be dangerous for the functioning of the cell. In addition, a more detailed study of the changes in mitophagy, mitochondrial biogenesis, and mitochondrial transport and the impact that these changes have on the clinical manifestations of cardiovascular diseases, is of great importance. The reason, which has not been fully elucidated, for the strengthening or weakening of the process at a particular stage of pathogenesis, is an obstacle in the creation of drugs aimed at modulating the function of mitochondrial homeostasis proteins. The relationship consisted in the interaction between mitochondrial fission and fusion proteins and the resulting changes in these interactions during the development of cardiac dysfunction, also require further study. It is known that fusion, fission, and mitophagy are closely related to each other and form a common mitochondrial quality system. Regulatory proteins of one of the processes can influence another process, thus exerting indirect regulation, for example, the activity of DRP1 (regulator of division) [71] and OPA1 [72] (regulator of fusion) proteins is associated with the implementation of mitophagy. The division of mitophagy supplies the damaged mitochondria for destruction through mitophagy or for preservation, giving a kind of “second chance” by merging with a healthy mitochondrion [73]. A more detailed study of the interactions of protein regulators of these processes, may help to better understand the disturbances in mitochondrial dynamics that occur in CVD. A separate perspective is the consideration of other regulatory proteins of mitochondrial fusion and fission as potential therapeutic targets, in addition to Drp1 and Mfn2, such as Fis1, OPA1 and Mfn1. As shown in the previous section, repurposing drugs with demonstrated ability to modulate mitochondrial dynamics previously used to treat other diseases, is a useful strategy to bring this class of drugs into clinical practice more quickly.

## 7. Conclusions

For various cardiovascular diseases, a disruption of the processes of mitochondrial dynamics was noted, leading to dysfunction and death of cardiomyocytes. At the same time, for different diseases and even during the pathogenesis of one disease, there is a multidirectional change in the processes of mitochondrial dynamics. The clearest pattern is observed for the fission and fusion of mitochondria. For the first, an increase was noted in cardiovascular diseases, and for the second, a weakening. To date, there are therapeutic developments targeted at regulatory proteins of mitophagy, fission and fusion of mitochondria. The most promising target is Drp1, which is the initiator of mitochondrial division. Regarding it, we found the largest number of successful developments, including clinical trials.

## Figures and Tables

**Figure 1 ijms-23-14741-f001:**
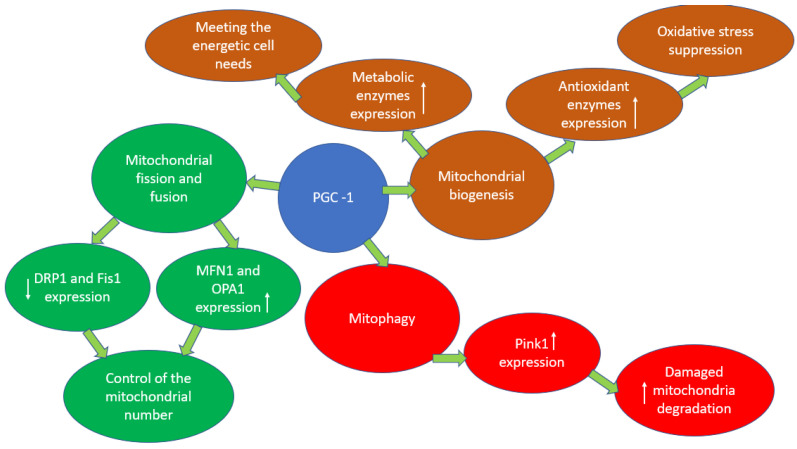
The multiple roles of PGC-1 α in the regulation of mitochondrial dynamics.

**Figure 2 ijms-23-14741-f002:**
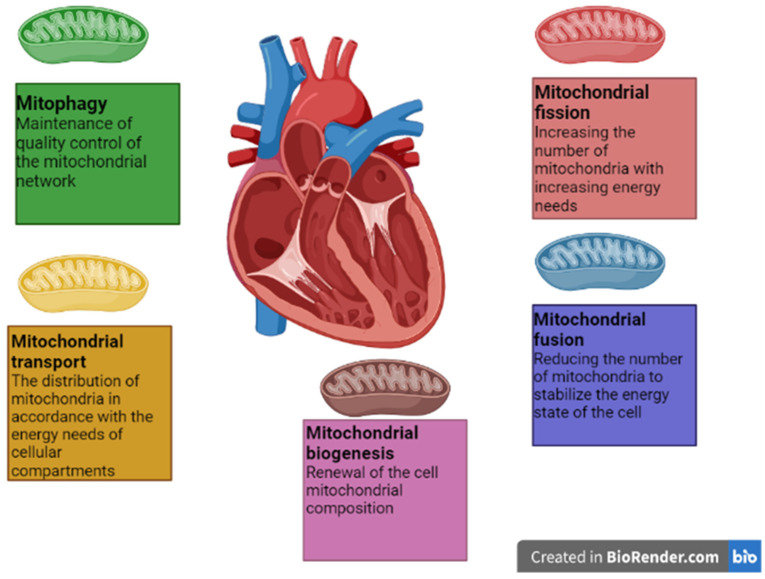
The scheme of mitochondrial dynamics in cardiomyocytes.

**Figure 3 ijms-23-14741-f003:**
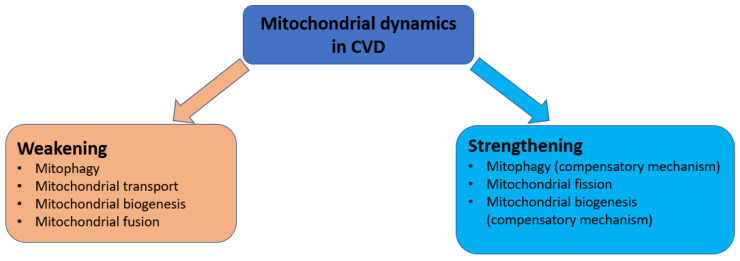
Mitochondrial dynamics disorders in cardiovascular diseases.

**Table 1 ijms-23-14741-t001:** Studies of mitochondrial dynamics disorders in CVD.

Disease	Model	Disruption of Mitochondrial Dynamics
Atherosclerosis	Mice, human cardiomyocyte culture	Decreased mitophagy
Heart failure	Mice, human cardiomyocyte culture	Decreased mitophagy
Ischemic heart disease	Mice	Increased mitophagy at the beginning of the IRI and decreased at the end of the IRI
Heart failure	Mice	Decreased mitochondrial transport
Mitochondrial cardiomyopathy	Human cardiomyocyte culture	Increased mitochondrial biogenesis
Hypertrophic cardiomyopathy	Human cardiomyocyte culture	Decreased mitochondrial biogenesis
Heart failure	Mice	Increased mitochondrial fission and decreased fusion
Metabolic cardiomyopathy	Rats	Increased mitochondrial fission and decreased fusion

**Table 2 ijms-23-14741-t002:** Modulation of mitochondrial dynamics for CVD treatment.

Therapy Compound	Action Mechanism	Study Type	Results
Liraglutide	Mitophagy strengthening through SIRT1 expression increasing	Animal model	Restoring myocardial function after a heart attack
Melatonin (1)	Activation of mitophagy	Animal models	Improving clinical symptoms of atherosclerosis and diabetic cardiomyopathy
Berberine	Activation of mitophagy	Animal model	Restoration of cardiac function in heart failure
mitoTEMPOL	Inhibition of mitophagy	Animal model	
mdivi-1	Suppression of the GTPase activity of DRP1	Animal model	Protection from ischemia/reperfusion and reduction in the possibility of myocardial infarction
P110 peptides	Inhibition of the DRP1 binding with Fis1	Animal model and cell model	Protection from septic cardiomyopathy development
Exenatide	Inhibition of the mitochondrial localization of Drp1	Cell model and clinical trial	Heart regeneration in cell model but no significant improvement in cardiac function in clinical trial
Resveratrol	Inhibition of the Drp1expression	Animal models and clinical trial	Improvement of clinical symptoms from patients with coronary heart disease
HO1	Increase in Mfn1/2 expression	Animal model	
Melatonin (2)	Increase in Mfn2 expression	Animal model	Attenuation of post-infarction injury

## Data Availability

Not applicable.

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
