# Peer review of "Targeting Mitochondrial Dynamics Proteins for the Development of Therapies for Cardiovascular Diseases"

_ijms, 2022, doi:10.3390/ijms232314741_

Round 1

Reviewer 1 Report

Comments to the author:

The manuscript entitled “Targeting Mitochondrial Dynamics Proteins for the Development of Therapies for Cardiovascular Diseases”. changes for each process of mitochondrial dynamics in cardiovascular diseases: fission and fusion of mitochondria, mitophagy, mitochondrial transport and biogenesis, and also analyze the prospects of the considered protein targets based on existing drug developments. It need major revisions.

Comments

1- Please specify role of PGC1 alpha in the mitochondrial dynamics

2- Authors provided only one figure, please include 2-3 more figures which will make manuscript easy for readers.

3- Role of AMPK and author protients should be elaborated

Author Response

Point 1: Please specify role of PGC1 alpha in the mitochondrial dynamics.

Response 1: The information was added in section 3.4.

Point 2: Authors provided only one figure, please include 2-3 more figures which will make manuscript easy for readers.

Response 2: The figures â„– 1 and 3 were added.

Point 3: Role of AMPK and author protients should be elaborated

Response 3: The information about AMPK was added in section 3.4. I don’t quite understand about “author protients” please clarify if it is nessesary.

Reviewer 2 Report

The authors reviewed the role of mitochondrial dynamics in the cardiovascular system. The potential relevance of proteins involved in mitochondrial dynamics as therapeutic targets was also discussed.

The review is interesting and well structured. I have few comments that should be addressed:

1) Mitophagy. Authors should begin this section with a brief description of autophagy and its relevance to the cardiovascular system. Furthermore, the authors should briefly mention the characterization of alternative forms of mitophagy.

2) The authors reviewed the relevant literature on important studies highlighting the role of mitochondrial dynamics and mitophagy in various models of cardiovascular diseases. It is sometimes difficult to understand in which model (in vivo, in vitro, patients) the results were observed. Please check the entire manuscript. A table, similar to table 1, can also be included.

3) Several experimental evidence suggests an interaction between mitochondrial fission and fusion with mitophagy. This should be briefly discussed.

4) The role of mitochondrial dynamics at baseline should be discussed. Furthermore, the authors should provide more details on mitochondrial morphology during fusion and fission.

Author Response

Point 1: Mitophagy. Authors should begin this section with a brief description of autophagy and its relevance to the cardiovascular system. Furthermore, the authors should briefly mention the characterization of alternative forms of mitophagy.

Response 1: The information was added in section 3.3.

Point 2: The authors reviewed the relevant literature on important studies highlighting the role of mitochondrial dynamics and mitophagy in various models of cardiovascular diseases. It is sometimes difficult to understand in which model (in vivo, in vitro, patients) the results were observed. Please check the entire manuscript. A table, similar to table 1, can also be included.

Response 2: The new table 1 was added.

Point 3: Several experimental evidence suggests an interaction between mitochondrial fission and fusion with mitophagy. This should be briefly discussed.

Response 3: The information was added in Discussion.

Point 4: The role of mitochondrial dynamics at baseline should be discussed. Furthermore, the authors should provide more details on mitochondrial morphology during fusion and fission.

Response 4: The information was added in section 3.

Round 2

Reviewer 1 Report

The manuscript can be accepted in its current form

Reviewer 2 Report

No further comments